# How Do Images Align and Complement LiDAR?
# Towards a Harmonized Multi-modal 3D Panoptic Segmentation

Yining Pan [1]  Qiongjie Cui [1]  Xulei Yang [2]  Na Zhao [1]

## Abstract

LiDAR-based 3D panoptic segmentation often struggles with the inherent sparsity of data from LiDAR sensors, which makes it challenging to accurately recognize distant or small objects. Recently, a few studies have sought to overcome this challenge by integrating LiDAR inputs with camera images, leveraging the rich and dense texture information provided by the latter. While these approaches have shown promising results, they still face challenges, such as misalignment during data augmentation and the reliance on post-processing steps. To address these issues, we propose **I**mage-**A**ssists-**L**iDAR (**IAL**), a novel multi-modal 3D panoptic segmentation framework. In IAL, we first introduce a modality-synchronized data augmentation strategy, PieAug, to ensure alignment between LiDAR and image inputs from the start. Next, we adopt a transformer decoder to directly predict panoptic segmentation results. To effectively fuse LiDAR and image features into tokens for the decoder, we design a Geometric-guided Token Fusion (GTF) module. Additionally, we leverage the complementary strengths of each modality as priors for query initialization through a Prior-based Query Generation (PQG) module, enhancing the decoder's ability to generate accurate instance masks. Our IAL framework achieves state-of-the-art performance compared to previous multi-modal 3D panoptic segmentation methods on two widely used benchmarks. Code and models are publicly available at https://github.com/IMPL-Lab/IAL.git.

[1]Singapore University of Technology and Design (SUTD) [2]Institute for Infocomm Research (I2R), A*STAR, Singapore. Correspondence to: Na Zhao <na_zhao@sutd.edu.sg>.

*Proceedings of the 42$^{nd}$ International Conference on Machine Learning*, Vancouver, Canada. PMLR 267, 2025. Copyright 2025 by the author(s).

## 1. Introduction

3D Panoptic segmentation simultaneously assigns semantic labels and identifies distinct instances, effectively unifying 3D semantic (Zhao et al., 2021; Xu et al., 2023b) and instance (Li & Zhao, 2024) segmentation to provide a holistic understanding of the scene. This task is particularly crucial for real-world applications, such as dynamic object tracking (Yang et al., 2023) and autonomous driving (Hong et al., 2021; Cao et al., 2024). LiDAR is an indispensable sensor for perceiving the 3D world, with its LiDAR point cloud typically serving as the sole input for 3D panoptic segmentation (Razani et al., 2021; Zhou et al., 2021; Li et al., 2022a). However, LiDAR inherently faces limitations in detecting small or distant objects due to its radial emission pattern, which results in sparse returns along each laser ray (Li et al., 2022b). Consequently, entities that are small or located at a distance may not receive sufficient information. In contrast, camera images provide denser and more detailed representations, effectively compensating for the sparsity in LiDAR data, particularly in these challenging scenarios.

This complementary nature has motivated the use of multi-modal information for enhanced panoptic segmentation. Recently, LCPS (Zhang et al., 2023) and Panoptic-FusionNet (Song et al., 2024) have pioneered LiDAR-and-image fusion methods for multi-modal 3D panoptic segmentation. However, these methods only perform augmentation on the LiDAR side, leading to misalignment between the two modalities. This misalignment hinders effective integration of information from both modalities, causing the models to rely predominantly on LiDAR data, rather than fully utilizing both LiDAR and image data. Moreover, the prediction heads in LCPS and Panoptic-FusionNet do not directly predict 3D panoptic segmentation results. Instead, they use a post-processing strategy that involves clustering instances after semantic segmentation (Zhou et al., 2021). This strategy presents two issues: 1) The post-processing step is inefficient and limits the effectiveness of segmentation by relying on preliminary results; 2) Their convolution-based prediction heads rely on local context, which may be suboptimal for panoptic segmentation as it requires global context for accurate predictions.

To address the first limitation, we propose a modality-

synchronized augmentation strategy – **PieAug**. PieAug ensures that the augmentation of multi-view images is synchronized with the augmentation of their corresponding LiDAR pairs, enabling well-aligned and enriched LiDAR and image inputs. Notably, PieAug is a general multi-modal data augmentation strategy designed for outdoor segmentation tasks. Its LiDAR-specific augmentation can be seen as a generalization of existing point cloud augmentation techniques, e.g., instance augmentation (Zhou et al., 2021), PolarMix (Xiao et al., 2022), LaserMix (Kong et al., 2023b).

To overcome the second post-processing limitation, we propose adopting a transformer decoder for multi-modal 3D panoptic segmentation, inspired by its success in 3D panoptic segmentation (Xiao et al., 2025) and multi-modal 3D object detection (Bai et al., 2022; Yan et al., 2023). By leveraging global context and directly predicting class labels and mask outputs, the transformer decoder eliminates the inefficiencies and constraints associated with post-processing. Despite its promise, adopting a transformer decoder introduces new challenges, particularly in *designing effective queries and tokens as inputs*. To overcome these challenges, we introduce a **Geometric-guided Token Fusion** (GTF) module and a **Prior-based Query Generation** (PQG) module. Combined with the PieAug strategy, these components form our proposed solution: a novel **Image-Assist-LiDAR** transformer-based framework, named **IAL**, for multi-modal 3D panoptic segmentation in autonomous driving scenarios.

GTF module integrates the sparse, cylinder-shaped LiDAR features with the compact, grid-shaped image features to create input tokens. Specifically, we adopt Cylinder3D (Zhu et al., 2020) to extract LiDAR features and use raw LiDAR points as geometric guidance to locate corresponding image patches for each cylindrical voxel. Additionally, we design a scale-aware positional embedding to encode the cylindrical voxels' locations and their receptive fields, facilitating the fusion of image patches and cylindrical voxels. This approach enhances feature fusion while mitigating projection errors caused by variations in cylindrical voxel shapes.

Our PQG module leverages prior knowledge from LiDAR and image inputs, which provide complementary strengths for object perception, to improve query initialization. Specifically, we generate two groups of instance queries – *geometric-prior* and *texture-prior* instance queries – derived from LiDAR and image modalities, respectively. Geometric-prior queries exploit LiDAR's geometric features, which are well-suited for detecting nearby or large objects rich in geometric information. In contrast, texture-prior queries leverage images by applying state-of-the-art detection and segmentation models, such as Grounding-DINO (Liu et al., 2023a) and SAM (Kirillov et al., 2023), to better identify distant and small objects. To handle challenging scenarios where both LiDAR and images fail to provide

reliable instance queries, we introduce a set of learnable parameters as *no-prior instance queries*. Consequently, the three groups of instance queries, combined with a set of semantic queries, are input into the transformer decoder to predict instance masks and semantic labels.

Our contributions can be summarized as: **1)** We present IAL, a novel transformer-based multi-modal framework for multi-modal 3D panoptic segmentation, eliminating the cumbersome post-processing steps required by previous methods. **2)** We propose PieAug, a multi-modal augmentation technique that not only addresses the asynchronization issue but also serves as a generalized formulation of existing LiDAR augmentation methods. **3)** We design the GTF and PQG modules that can effectively fuse image and LiDAR features as tokens and queries for the transformer decoder. **4)** Our IAL achieves state-of-the-art performance in outdoor panoptic segmentation, surpassing previous methods by 2.5% and 4.1% in PQ on the nuScenes and SemanticKITTI benchmarks, respectively.

## 2. Related Work

**3D Panoptic Segmentation.** Most advanced approaches for LiDAR-based 3D panoptic segmentation can be categorized into three main groups: top-down, bottom-up, and single-path methods (Li & Chen, 2022). The top-down method typically follows a detection-first principle, where bounding boxes are predicted initially, followed by the generation of instance masks from points within those boxes (Sirohi et al., 2021; Xu et al., 2023a; Ye et al., 2023). In contrast, bottom-up methods (Zhou et al., 2021; Hong et al., 2021; Razani et al., 2021; Xu et al., 2022; Li et al., 2022a) begin with semantic segmentation predictions and then generate instance masks through operations such as grouping and clustering. Both top-down and bottom-up approaches are limited by the performance of their preliminary predictions (object detection or semantic segmentation, respectively), which hinders their ability to achieve holistic perception. On the other hand, single-path methods treat panoptic segmentation as a unified task, simultaneously addressing the segmentation of both "stuff" and "thing" classes. For example, MaskRange (Gu et al., 2022), MaskPLS (Marcuzzi et al., 2023), and P3Former (Xiao et al., 2025) utilize learnable queries to predict masks and classes for both "thing" and "stuff" object types. Additionally, PUPS (Su et al., 2023) and DQFormer (Yang et al., 2025) explore the use of auxiliary categorical and positional embeddings for query initialization. However, these methods mainly rely on LiDAR. inputs and often struggle with recognizing small or distant objects, where geometric information degrades. This limitation inspires the use of camera images, which contain rich texture information, to enhance panoptic segmentation.

**Multi-modal 3D Scene Understanding.** Multi-modal

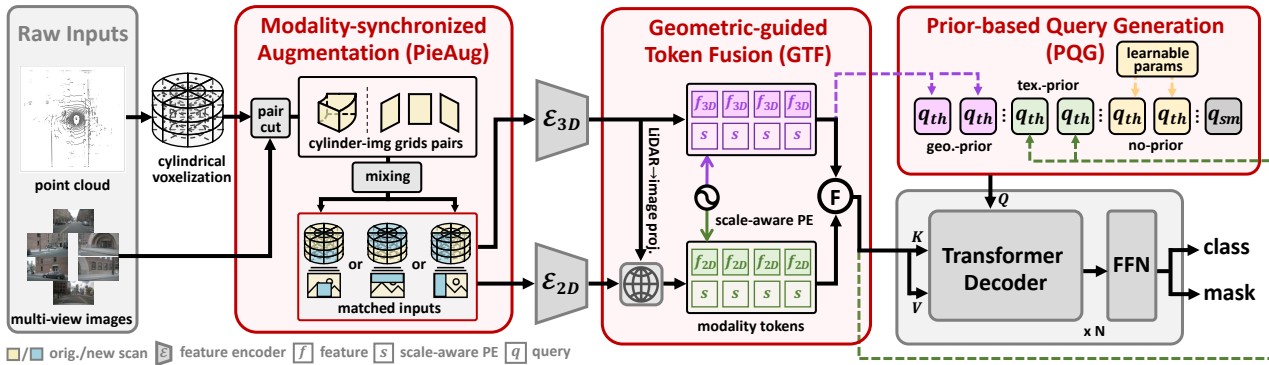

Figure 1. **The architecture overview of our Image-Assists-LiDAR (IAL) framework**. We first voxelize the point cloud into cylindrical voxels. In **PieAug**, we synchronize augmentation by pairing cylindrical and image grids, mixing original and new scans in different modes. Using a transformer-based structure, we design token fusion (GTF) and query initialization (PQG) modules to align and complement both modalities. In **GTF**, features from both modalities are bridged via LiDAR-to-image projection and scale-aware positional embeddings. We generate instance queries for "thing" classes using modality knowledge (specifically, geometric and texture priors from LiDAR and image features) and learnable parameters (for instances without advanced modality priors) in **PQG**. These instance queries, along with semantic queries, are iteratively updated through multiple transformer decoder layers to produce the final panoptic predictions.

learning with LiDAR and images has been extensively studied for 3D semantic segmentation (Zhuang et al., 2021; Krispel et al., 2020; Yan et al., 2022; Li et al., 2023b; Liu et al., 2023b; Man et al., 2023; Wu et al., 2024; An et al., 2025) and object detection (Liang et al., 2022; Yin et al., 2024; Bai et al., 2022; Li et al., 2024). A key challenge in these tasks is to effectively fuse the features of both modalities to leverage their complementary strengths. To address this, existing methods design fusion modules to align data from different sensors. Additionally, some object detection studies (Bai et al., 2022; Chen et al., 2023; Li et al., 2024; Zhang et al., 2024) have shown that image-driven queries improve the detection of challenging objects. While these approaches are not directly applicable to panoptic segmentation, they inspire our design of the Geometric-guided Token Fusion (GTF) and Prior-based Query Generation (PQG). For multi-modal 3D panoptic segmentation, LCPS (Zhang et al., 2023) and Panoptic-FusionNet (Song et al., 2024) are two pioneering works. LCPS designs a point-to-mask mapping for LiDAR to image fusion, while Panoptic-FusionNet directly applies point-to-pixel mapping by geometric information. However, these methods primarily apply augmentations only to LiDAR, leading to misalignment between modalities and an over-reliance on LiDAR features. Additionally, their reliance on post-processing makes panoptic inference inefficient and limits the effectiveness of true multi-modal perception.

**LiDAR-Based Data Augmentation.** Studies on LiDAR-based data augmentation often rely on instance- or scene-level mixing. Instance mixing methods (Zhou et al., 2021; Xiao et al., 2022; Zhao et al., 2022) augment point clouds by copying instance points from one scan to another, while scene-wise mixing approaches, such as LaserMix (Kong et al., 2023b) and PolarMix (Xiao et al., 2022), divide the scene into multiple intervals along inclination or azimuth

angles and selectively swap these intervals between scans. RangeFormer (Kong et al., 2023a) represents the 3D scene as a range-view image and applies tailored augmentation strategies for range-view learning. Similarly, UniMix (Zhao et al., 2024) extends scene-wise mixing to different attributes, including intensity and semantic channels. In multi-modal 3D scene understanding, data augmentation for both modalities remains underexplored. LaserMix++ (Kong et al., 2024) extends LaserMix to multi-modal scenes but relies on a single augmentation strategy. MSeg3D (Li et al., 2023b) applies asymmetric augmentation to each modality, using only simple local transformations – reducing the risk of misalignment but at the cost of data diversity. To address these limitations, we propose a general multi-modal augmentation strategy that incorporates diverse instance- and scene-level mixing to enhance both cross-modal alignment and panoptic segmentation performance.

## 3. Methodology

In the multi-modal 3D panoptic segmentation, we are given a 3D point cloud consisting of $N$ discrete sampling points, denoted as $\mathbf{P} = \{\mathbf{p}_j \in \mathbb{R}^{1\times4}\}_{j=1}^N$, where each point $\mathbf{p}_j$ contains its Cartesian coordinates in Euclidean space and its reflection intensity. The point cloud is associated with $K$ view images, represented as $\mathcal{I} = \{\mathbf{I}_k \in \mathbb{R}^{H\times W\times3}\}_{k=1}^K$, $H$ and $W$ denote the height and width of images. The goal of this task is to effectively utilize $\mathbf{P}$ and $\mathcal{I}$ to predict both semantic and instance labels for each point.

**Framework Overview.** In this paper, we introduce Image-Assist-LiDAR (IAL), a novel transformer-based framework for multi-modal 3D panoptic segmentation, as illustrated in Fig. 1. To process the sparse and irregular LiDAR point cloud, we first apply *cylindrical voxelization*, converting points into cylindrical-shaped voxels based on their polar coordinates. As a result, each voxel $\mathbf{v}_i$ contains a varying

number of points, with voxel shapes differing along the radial axis. The point cloud $\mathbf{P}$ can then be represented as $\mathbf{V} = \{\mathbf{v}_i\}_{i=1}^M$, where $M$ is the number of valid cylindrical voxels. We apply modality-synchronized augmentation through our proposed PieAug strategy (Sec. 3.1), ensuring consistency across LiDAR and image data by pairing each voxel with its corresponding image regions and employing a generalized augmentation operator for diverse effects.

The augmented 3D voxels and images are then processed by 3D encoder $\mathcal{E}_{3D}$ and 2D encoder $\mathcal{E}_{2D}$, extracting voxel-wise features $\mathbf{F}^{3D} \in \mathbb{R}^{M \times D}$ and image features $\mathbf{F}^{2D} \in \mathbb{R}^{K \times H \times W \times D}$, where $D$ is the feature dimension. The 3D encoder uses Cylinder3D (Zhu et al., 2020), known for its strong generalization in 3D panoptic segmentation (Xiao et al., 2025; Zhang et al., 2023), and 2D encoder is Swift-Net (Oršić & Šegvić, 2021) with a ResNet-18 backbone.

Next, we use $\mathbf{F}^{3D}$ and $\mathbf{F}^{2D}$ to create tokens and queries for a transformer decoder, enabling cross-modal interaction. Token features are formed by concatenating voxel features with their aligned image counterparts, guided by a unified, scale-aware positional embedding (Sec.3.2). Meanwhile, instance queries $\mathbf{q}_{th}$ are initialized using modality-compensated priors (Sec.3.3). These queries, along with semantic queries $\mathbf{q}_{sm}$ are fed into the transformer decoder to predicts the instance masks and semantic labels.

## 3.1. Modality-Synchronized Augmentation

To mitigate modality misalignment and enhance diversity during data augmentation, we propose PieAug. The key idea is to extract a flexible number of cylindrical voxels along the height, angle, or radius axes – analogous to cutting a variable-sized pie slice from a cake – and swap it with a corresponding slice from another scene, as illustrated in Fig. 2. To maintain synchronization across modalities, each 3D "pie" is paired with its corresponding image patch, which is exchanged simultaneously.

For simplicity, we illustrate the process of finding the corresponding image patch for a single voxel from one camera view. This can be easily extended to a pie-shaped region with multiple image views. Given a voxel containing $N_i$ points, denoted as $\mathbf{v}_i = \{\mathbf{p}_j\}_i^{N_i}$, where $\sum_{i=1}^M N_i = N$, we first project the LiDAR point $\mathbf{p}_j$ from 3D coordinate system to its corresponding 2D coordinates in the image plane using the following transformation:

$$\pi(\mathbf{p}_j) = \mathbf{K} \times \mathbf{T} \times [\mathbf{p}_{j,1}, \mathbf{p}_{j,2}, \mathbf{p}_{j,3}, 1]^\top, \quad (1)$$

where $\mathbf{K} \in \mathbb{R}^{3 \times 3}$ is the camera intrinsic matrix and $\mathbf{T} \in \mathbb{R}^{4 \times 4}$ is the extrinsic transformation matrix. Next, we define $\mathbf{g}_i$ as the bounding rectangle that encloses all projected points $\pi(\mathbf{p}_j)$. This ensures that each voxel corresponds to a specific region in the image, denoted as $\langle \mathbf{v}_i, \mathbf{g}_i \rangle$:

$$\mathbf{g}_i = \mathcal{B}\left(\{\pi(\mathbf{p}_j) \mid \mathbf{p}_j \in \mathbf{v}_i\}\right). \quad (2)$$

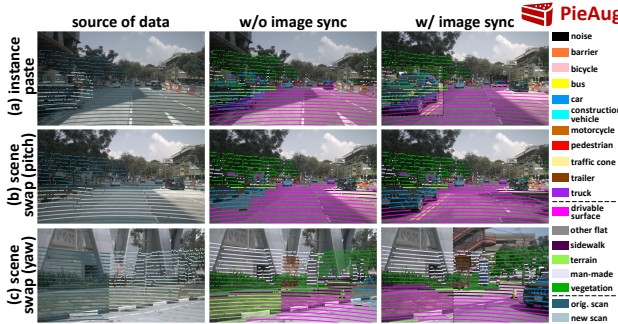

*Figure 2.* Motivation and implementation variants of **PieAug**. Each column illustrates the motivation for LiDAR-image synchronized augmentation. Each row displays a different *pie-cut* strategy. Point clouds are projected on camera images, with colors indicating semantic labels or data sources. Best viewed in color.

Here $\mathcal{B}(\cdot)$ is the operator that fits a bounding rectangle enclosing a set of pixels.

During augmentation, we determine the size and position of the *pie-cut* using a 3D binary mask $\mathbf{S} \in \{0, 1\}^{R \times \Theta \times Z}$, where $R$, $\Theta$, and $Z$ represent the binning resolution along the radial, angular, and height axes of the cylindrical voxelization, respectively. Each voxel is assigned a corresponding mask value $\mathbf{S}(r, \theta, z)$, where $\mathbf{S}(r, \theta, z) = 1$ indicates that the voxel is replaced with one from a new scan; otherwise, it remains unchanged from the original scene. Consequently, the augmented cylinder $\mathbf{V}^{aug}$ is obtained as follows:

$$\mathbf{V}^{aug} = \mathbf{V}^{org} \otimes (1 - \mathbf{S}) + \mathbf{V}^{new} \otimes \mathbf{S}, \quad (3)$$

where $\otimes$ denotes an element-wise masking operation, multiplying the voxel values by the corresponding mask. Since each voxel $\mathbf{v}_i$ aligns with an image region $\mathbf{g}_i$, we apply image augmentation in parallel using the same mask $\mathbf{S}$.

**Instance Pasting.** We first illustrate how PieAug achieves instance-level augmentation (copy and paste) by selecting pie-cut voxels corresponding to $s$ sampled instances from a new scene. We apply transformations such as translation, rotation, and scaling to each instance. Next, we identify the indices of voxels that overlap with the $s$ transformed instances as $\mathcal{C}$. The mask $\mathbf{S}$ is then constructed as:

$$\mathbf{S} = \bigcup_{r=1, \theta=1, z=1}^{R \times \Theta \times Z} \mathbb{1}[(r, \theta, z) \in \mathcal{C}]. \quad (4)$$

**Scene Swapping.** We then illustrate how PieAug achieves scene-level augmentation, including scene swapping, which involves dividing the voxels evenly along a chosen axis (height or angle) and swapping them alternately. We achieve this by selecting all voxels along the radial axis, as well as one of the height or angle axes, and then freely choosing a number of slices from the remaining axis. For example, the mask for selecting $b$ slices from the angle axis is defined as:

$$\mathbf{S}(r, \theta, z) = \begin{cases} 1, & \text{if } \theta \in \mathcal{O} \\ 0, & \text{otherwise} \end{cases} \quad (5)$$

Here, $\mathcal{O}$ denotes the set of indices corresponding to the selected $b$ angle slices. A similar masking strategy can be applied to height slices.

**Remarks.** With the instance-level and scene-level augmentation capabilities described above, PieAug generalizes most existing LiDAR-based augmentation techniques. For example: 1) Panoptic-PolarNet: perform instance-level augmentation by sampling instances based on their semantic labels and applying transformations using translation and XY-plane rotation. 2) PolarMix (instance branch): perform instance-level augmentation by selecting instances based on their labels and applying transformations by rotating duplicated instances multiple times along the Z-axis. 3) PolarMix (scene branch): perform scene-level augmentation by selecting half of the slices along the azimuth angle. 4) LaserMix: perform scene-level augmentation by selecting slices at different inclination angles. As a result, PieAug offers greater flexibility in combining instance-level and scene-level augmentations by performing multiple augmentation rounds with different voxel indices. Furthermore, due to the synchronized transformations applied to both modalities, PieAug ensures well-aligned LiDAR and image augmentations.

### 3.2. Geometric-Guided Token Fusion

Since cylindrical voxelization results in voxels of varying sizes (larger in regions farther from the central origin), this poses two key challenges for generating multi-modal tokens: 1) how to align image features with LiDAR features, and 2) how to effectively fuse. To address these issues, we propose **Geometric-guided Token Fusion (GTF)**, as illustrated in Fig. 3, which leverages the rich geometric information from LiDAR to guide alignment and enable effective fusion.

Specifically, we align features at the voxel level by projecting all physical points within a voxel $\mathbf{v}_i$ onto the image plane and averaging their corresponding image features to create an aggregated representation:

$$\tilde{\mathbf{F}}_i^{2D} = \frac{1}{N_i} \sum_j^{N_i} \mathbf{F}^{2D}(\pi(\mathbf{p}_j)). \qquad (6)$$

We refer to the paired voxel-wise LiDAR feature $\mathbf{F}_i^{3D}$ and image feature $\tilde{\mathbf{F}}_i^{2D}$ as the *contents* of the $i$-th multi-modal token. Notably, aggregating image features by projecting all physical points within a voxel preserves feature validity. As shown in Fig. 3(a), using only the voxel centroid can lead to misalignment, as the projected location may not correspond accurately to the relevant image region.

Position embedding (PE) has proven effective in aligning features from different modalities (Yan et al., 2023). However, due to the varying sizes of cylindrical voxels, encoding only the voxel centroid is suboptimal, as a single point may not sufficiently represent the voxel's perceptive field. Even when using physical points for PE, capturing the full perceptive field of a voxel or its corresponding image region

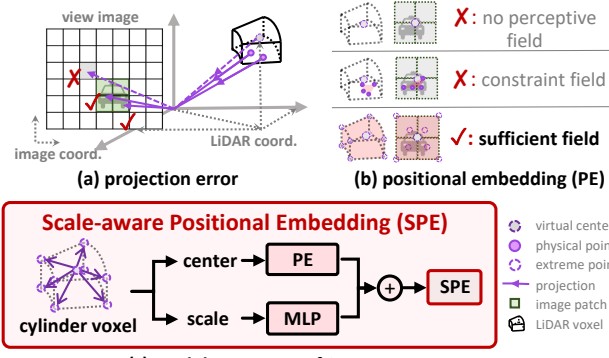

**(a) projection error**  **(b) positional embedding (PE)**

**(c) module structure of SPE**

*Figure 3.* **(a)** and **(b)** illustrate two challenges in LiDAR–image fusion introduced by cylindrical voxelization. In (a), relying on a virtual voxel center can lead to projection errors, which become more pronounced for larger voxels and cause mismatches between LiDAR and image data. This motivates the use of physical points. In (b), canonical PE (row 1) overlooks the varying sizes of voxel and image features, while focusing solely on the region of physical points (row 2) limits the receptive field. Hence, we introduce a **scale-aware PE** that uses extreme points, as shown in **(c)**.

remains inadequate, as illustrated by the red region in the second row of Fig. 3(b). This limitation arises because physical points provide only a partial representation of objects in both the LiDAR and image domains, especially at greater distances where LiDAR points are sparsely distributed. To address this issue, we propose a unified **Scale-aware Position Embedding (SPE)** for both LiDAR and image tokens. SPE ensures consistent perception of perceptive fields by incorporating shared scale embeddings to both modalities, as shown in the third row of Fig. 3(c).

Specifically, as illustrated in Fig. 3(c), we introduce an *extreme point set*, denoted as $\hat{\mathbf{v}}_i = \{\hat{\mathbf{p}}_j\}_{j=1}^8$, consisting of the eight corners of each cylindrical voxel to represent its scale. In the LiDAR space, these extreme points define the spatial partitioning of the scene. In the image space, they outline potential perceptive field regions, particularly in areas with sparse LiDAR points, allowing for a more adaptive perception range. In SPE, each cylindrical voxel is embedded with both its scale and centroid position:

$$\mathbf{s}_i = \psi(\mathrm{Avg}(\hat{\mathbf{v}}_i)) + \phi(||\hat{\mathbf{v}}_i - \mathrm{Avg}(\hat{\mathbf{v}}_i)||_2), \qquad (7)$$

where $\mathrm{Avg}(\cdot)$ denote the averaging method and and $||\cdot||$ is L2 norm operation in Cartesian space. $\psi$ represents a mixed-parameterized positional encoding function (Xiao et al., 2025), which embeds the centroid position in both Cartesian and polar spaces. $\phi$ is a multi-layer perceptron (MLP) that projects the scale feature into the same dimension as the centroid embedding. With SPE, the final fused multi-modal token $\mathbf{F}_i^{fuse}$ for $i$-th cylindrical voxel is obtained by:

$$\mathbf{F}_i^{fuse} = \mathrm{Cat}[(\mathbf{F}_i^{3D} \oplus \mathbf{s}_i), (\tilde{\mathbf{F}}_i^{2D} \oplus \mathbf{s}_i)]. \qquad (8)$$

### 3.3. Prior-Based Query Generation

Previous works, such as P3Former (Xiao et al., 2025), initialize queries by a set of learnable parameters. However,

*Table 1.* Preliminary study of positional embedding for objects of thing classes. We conduct the experiment on our LiDAR branch. "GT" denotes using the ground truth center position, while "Noise" denotes adding Gaussian noise with a kernel size of 3 to the GT center position. "th" and "st" is the thing and stuff classes.

| Modality | GT | Noise | PQ | mIoU | $PQ^{th}$ | $PQ^{st}$ |
|---|---|---|---|---|---|---|
| LiDAR | | | 77.0 | 75.9 | 77.8 | 75.7 |
| LiDAR | ✓ | | 83.2 | 82.3 | 88.5 | 74.4 |
| LiDAR | ✓ | ✓ | 81.8 | 79.8 | 86.8 | 73.6 |

such queries tend to prioritize easier samples while neglecting more challenging ones. Additionally, preliminary results have validated that accurate positional embedding significantly enhances the model's ability to locate objects. As demonstrated in Table 1, applying the ground truth center position for thing classes resulted in a 6.2% improvement in overall PQ and a 10.7% improvement in PQ for thing classes specifically. Even adding noise on the position, the improvement is still significant.

Inspired by this observation, we propose the **Prior-based Query Generation (PQG)** module to explicitly leverage texture features from the image domain, and geometric information from LiDAR domain as prior knowledge to generate well-informed initializations for instance queries. Specifically, we design three groups of queries: geometric-prior, texture-prior, and no-prior queries.

**Geometric-Prior Query.** Potential geometric-prior instances are those for which LiDAR features exhibit minimal degradation, allowing geometric characteristics to provide sufficient information for accurate positional hints. Compared to the rich texture features captured in images, LiDAR features offer more precise location predictions. Therefore, we generate location hints for geometric-prior queries by predicting a center heatmap and performing Non-Maximum Suppression (NMS) sampling according to confidence scores and range radius threshold. Specifically, we predict the class-agnostic heatmap in the polar Bird's-Eye-View (BEV) space using a structure similar to (Yin et al., 2021; Bai et al., 2022). For each selected instance proposal (identified by its coordinates in the center heatmap), we lift it into 3D space by averaging all valid voxels across the height dimension.

**Texture-Prior Query.** For objects that are small or located far from the sensor, geometric information often becomes unreliable, making accurate location prediction difficult. To address this issue, we use texture information from images to discover potential texture-prior instances. First, we extract mask proposals using the pre-trained image segmentation models Grounding-DINO and SAM (Liu et al., 2023a; Kirillov et al., 2023). We then lift each 2D mask into a 3D frustum and collect all the 3D points that fall within it. To mitigate noise caused by overlapping along the depth dimension, we cluster these points into several groups using

the unsupervised DBSCAN (Ester et al., 1996) algorithm. Finally, the centroids of these clusters serve as location hints for texture-prior instances.

Given location hints from both modalities, we apply Farthest Point Sampling (FPS) to obtain a fixed number $l_{pr}$ of location hints. It is worth noting that for large objects easily recognizable by both LiDAR and images, global sampling provides a holistic view and reduces redundant candidate proposals. Similar to the positional embedding used for multi-modal tokens, we apply SPE to embed both the query location and scale features. We then extract the query content by indexing into the corresponding voxel features $\mathbf{F}^{fuse}$, and finally add the SPE to this content to form the final query representation.

**No-Prior Query.** We hypothesize that instances without advanced priors exhibit a specific feature representation paradigm, allowing them to be recognized through a set of learnable parameters. We set the number of no-prior queries as $l_{lt}$. This implicit paradigm learning enables the model to search within a smaller candidate pool and effectively identify potential instances.

All geometric-, texture-, and no-prior queries are concatenated and fed into the transformer decoder for "thing" class prediction, i.e., 3D instance segmentation. Semantic queries are initialized following the approach in P3Former, with auxiliary semantic supervision applied. These semantic queries are used to predict segmentation results. We follow the process in P3Former to combine the instance and semantic predictions into the final panoptic segmentation.

## 4. Experimental Results

### 4.1. Experimental Setting

**Datasts. nuScenes** (Caesar et al., 2020; Fong et al., 2022) is a large-scale, multi-modal dataset designed for autonomous driving, containing data from a 32-beam LiDAR, 5 radars, and 6 RGB cameras. It includes 40,157 frames of outdoor scenes, with 34,149 frames labeled for training and validation, and the remaining reserved for testing. The panoptic annotations cover 10 "thing" classes, 6 "stuff" classes, and 1 class for noisy labels. **SemanticKITTI** (Behley et al., 2019; 2021) is an outdoor dataset derived from KITTI Vision Benchmark (Geiger et al., 2012). It includes data from a 64-beam LiDAR sensor and two front-view cameras, including 8 "thing" classes and 11 "stuff" classes, comprising 19,130 frames for training, 4,071 frames for validation, and 20,351 frames for testing.

**Evaluation Metrics.** Consistent with the standard works (Kirillov et al., 2018; Zhang et al., 2023; Xiao et al., 2025), panoptic quality (PQ) is selected as the primary metric. PQ is defined as the product of segmentation quality

*Table 2.* **Comparison of panoptic segmentation performance on the nuScenes validation set.** Top results are shown in **bold**. "M." indicates which modality (or modalities) each method uses. "P.-PCSCNet" is the LiDAR branch of "P.-FusionNet" (Song et al., 2024).

| Method | M. | PQ | PQ$^\dagger$ | RQ | SQ | PQ$^{th}$ | RQ$^{th}$ | SQ$^{th}$ | PQ$^{st}$ | RQ$^{st}$ | SQ$^{st}$ | mIoU |
|---|---|---|---|---|---|---|---|---|---|---|---|---|
| DS-Net (Hong et al., 2021) | L | 42.5 | 51.0 | 50.3 | 83.6 | 32.5 | 38.3 | 83.1 | 59.2 | 70.3 | 84.4 | 70.7 |
| EfficientLPS (Sirohi et al., 2021) | L | 62.0 | 65.6 | 73.9 | 83.4 | 56.8 | 68.0 | 83.2 | 70.6 | 83.6 | 83.8 | 65.6 |
| P.-PolarNet (Zhou et al., 2021) | L | 67.7 | 71.0 | 78.1 | 86.0 | 65.2 | 74.0 | 87.2 | 71.9 | 84.9 | 83.9 | 69.3 |
| P.-PHNet (Li et al., 2022a) | L | 74.7 | 77.7 | 84.2 | 88.2 | 74.0 | 82.5 | 89.0 | 75.9 | 86.9 | 86.8 | 79.7 |
| CFNet (Li et al., 2023c) | L | 75.1 | 78.0 | 84.6 | 88.8 | 74.8 | 82.9 | 89.8 | 76.6 | 87.3 | 87.1 | 79.3 |
| CenterLPS (Mei et al., 2023) | L | 76.4 | 79.2 | 88.0 | 86.2 | 77.5 | 88.4 | 87.1 | 74.6 | 87.3 | 84.9 | 77.1 |
| LCPS (Zhang et al., 2023) | L | 72.9 | 77.6 | 82.0 | 88.4 | 72.8 | 80.5 | 90.1 | 73.0 | 84.5 | 85.5 | 75.1 |
| P.-PCSCNet (Song et al., 2024) | L | 72.7 | 75.4 | 84.8 | 86.4 | 71.2 | 82.9 | 86.6 | 75.1 | 84.2 | 84.2 | 69.8 |
| P3Former (Xiao et al., 2025) | L | 75.9 | 78.9 | 84.7 | 89.7 | 76.9 | 83.3 | 92.0 | 75.4 | 87.1 | 86.0 | 76.8 |
| IAL (our LiDAR branch) | L | 77.0 | 79.6 | 85.1 | 90.2 | 77.8 | 83.8 | 92.6 | 75.7 | 87.3 | 86.2 | 75.9 |
| LCPS (Zhang et al., 2023) | L+C | 79.8 | 84.0 | 88.5 | 89.8 | 82.3 | 89.6 | 91.7 | 75.6 | 86.5 | 86.7 | 80.5 |
| P.-FusionNet (Song et al., 2024) | L+C | 77.2 | 79.3 | 87.2 | 87.8 | 77.5 | 87.7 | 88.2 | 76.2 | 85.9 | 86.0 | 73.4 |
| **IAL (ours)** | L+C | **82.3** | **84.7** | **89.7** | **91.5** | **85.3** | **90.6** | **94.1** | **77.3** | **88.2** | **87.2** | **80.6** |

*Table 3.* **Comparison on the nuScenes test set.** Top and runner-up results are marked in **bold** and underline, respectively. "*" indicates the use of additional temporal frames and detection annotations. Our method is evaluated without test-time augmentation or ensembling.

| Method | M. | PQ | PQ$^\dagger$ | RQ | SQ | PQ$^{th}$ | RQ$^{th}$ | SQ$^{th}$ | PQ$^{st}$ | RQ$^{st}$ | SQ$^{st}$ | mIoU |
|---|---|---|---|---|---|---|---|---|---|---|---|---|
| EfficientLPS (Sirohi et al., 2021) | L | 62.4 | 66.0 | 74.1 | 83.7 | 57.2 | 68.2 | 83.6 | 71.1 | 84.0 | 83.8 | 66.7 |
| P.-PolarNet (Zhou et al., 2021) | L | 63.6 | 67.1 | 75.1 | 84.3 | 59.0 | 69.8 | 84.3 | 71.3 | 83.9 | 84.2 | 67.0 |
| P.-PHNet (Li et al., 2022a) | L | 80.1 | 82.8 | 87.6 | 91.1 | 82.1 | 88.1 | 93.0 | 76.6 | 86.6 | 87.9 | 80.2 |
| CPSeg (Li et al., 2023a) | L | 73.2 | 76.3 | 82.7 | 88.1 | 72.9 | 81.3 | 89.2 | 74.0 | 85.0 | 86.3 | 73.7 |
| MaskPLS (Marcuzzi et al., 2023) | L | 61.1 | 64.3 | 68.5 | 86.8 | 54.3 | 58.8 | 87.8 | 72.4 | 84.5 | 85.1 | 74.8 |
| LCPS (Zhang et al., 2023) | L | 72.8 | 76.3 | 81.7 | 88.6 | 72.4 | 80.0 | 90.2 | 73.5 | 84.6 | 86.1 | 74.8 |
| LidarMultiNet* (Ye et al., 2023) | L | 81.4 | 84.0 | 88.9 | 91.3 | 83.9 | 89.9 | 93.1 | 77.3 | 87.1 | **88.2** | **82.2** |
| IAL (our LiDAR branch) | L | 75.1 | 77.7 | 83.0 | 90.1 | 75.0 | 80.9 | 92.4 | 75.2 | 86.5 | 86.4 | 73.3 |
| 4DFormer (Athar et al., 2023) | L+C | 78.0 | 81.4 | 86.6 | 89.7 | 80.0 | 87.8 | 90.9 | 74.6 | 84.5 | 87.6 | 80.4 |
| LCPS (Zhang et al., 2023) | L+C | 79.5 | 82.3 | 87.7 | 90.3 | 81.7 | 88.6 | 92.2 | 75.9 | 86.3 | 87.3 | 78.9 |
| **IAL (ours)** | L+C | **82.0** | **84.3** | **89.3** | **91.6** | **84.8** | **90.2** | **93.8** | **77.5** | **87.8** | 87.8 | 79.9 |

(SQ) and recognition quality (RQ):

$$PQ = \underbrace{\frac{\sum_{TP} IoU}{|TP|}}_{SQ} \times \underbrace{\frac{|TP|}{|TP| + \frac{1}{2}|FP| + \frac{1}{2}|FN|}}_{RQ}, \quad (9)$$

where IoU denotes the Intersection over Union, TP denotes True Positives and so as for others. Theses metrics can be further extended to "thing" and "stuff" classes, denoted as PQ$^{th}$, PQ$^{st}$, RQ$^{th}$, RQ$^{st}$, SQ$^{th}$, and SQ$^{st}$. We also report PQ$^\dagger$ (Porzi et al., 2019), which replaces PQ with mIoU for stuff classes.

**Implementation Details.** We follow standard practice (Zhou et al., 2021; Li et al., 2022a; Xiao et al., 2025) to represent point clouds by discretizing the 3D space into cylindrical voxels of size $[480 \times 360 \times 32]$. The LiDAR branch is built upon the architecture of P3former. For nuScenes, the polar coordinate range is defined as $[-50m, 50m] \times [0, 2\pi] \times [-5m, 3m]$, while for SemanticKITTI, the height range is adjusted to $[-4m, 2m]$. All images are resized to $640 \times 360$. For augmentation, we employ the following strategies: instance pasting and scene-swapping (split the scene along the height and angle axes, with the number of splits randomly chosen from $[3, 4, 5]$ each time). We set the ratio of application for these three augmentation

strategies to 0.4:0.05:0.05, respectively. Additionally, we perform basic transformations including random rotation, flipping, and scaling. We set the number of prior-based and no-prior instance queries to $l_{pr} = l_{lt} = 128$. We use AdamW (Kingma & Ba, 2014) as the optimizer, with a default weight decay of 0.01. The entire model is trained from scratch with a batch size of 2, using 4 NVIDIA A40 GPUs. The training spans 80 epochs for nuScenes and 36 epochs for SemanticKITTI. The initial learning rate is set to 0.0008 and decays by half at epochs [60,75] for nuScenes and [30,32] for SemanticKITTI, respectively. All model results listed in the following sections are NOT employed with any test-time augmentation (TTA) method.

### 4.2. Benchmark Results

**nuScenes**. We present comprehensive comparison results for LiDAR panoptic segmentation performance on the nuScenes validation and test sets, as shown in Table 2 and Table 3. Due to the limited number of multi-modal methods, currently only LCPS (Zhang et al., 2023) and Panoptic-FusionNet (Song et al., 2024), we also include LiDAR-only methods for comparison. Notably, our method IAL achieves the best performance across all metrics on the validation set and ranks first or second on most metrics

*Table 4.* Comparison of panoptic segmentation performance on the SemanticKITTI validation set. Top results are shown in **bold**.

| Method | M. | PQ | PQ$^\dagger$ | RQ | SQ | mIoU |
|---|---|---|---|---|---|---|
| P.-PolarNet | L | 59.1 | 64.1 | 70.2 | 78.3 | 64.5 |
| DS-Net | L | 57.7 | 63.4 | 68.0 | 77.6 | 63.5 |
| EfficientLPS | L | 59.2 | 65.1 | 69.8 | 75.0 | 64.9 |
| P.-PHNet | L | 61.7 | – | – | – | 65.7 |
| CenterLPS | L | 62.1 | 67.0 | 72.0 | 80.7 | – |
| LCPS | L | 55.7 | 65.2 | 65.8 | 74.0 | 61.1 |
| P3Former | L | 62.6 | 66.2 | 72.4 | 76.2 | – |
| IAL (LiDAR) | L | 62.0 | 65.1 | 71.9 | 76.0 | 64.9 |
| LCPS | L+C | 59.0 | **68.8** | 68.9 | 79.8 | 63.2 |
| **IAL (ours)** | L+C | **63.1** | 66.3 | **72.9** | **81.4** | **66.0** |

in the test set. Specifically, IAL outperforms LCPS and Panoptic-FusionNet by a significant margin of 2.5% and 5.1% in PQ on the validation set, as shown in Table 2. This improvement is attributed to superior performance in both recognition (surpassing the two previous works by 1.2% and 2.5% in RQ, respectively) and segmentation (surpassing them by 1.7% and 3.7% in SQ). Furthermore, our model demonstrates superior performance on both "thing" and "stuff" classes, achieving a 7.8% and 1.1% improvement in metrics compared to the latest work, Panoptic-FusionNet. Compared to the LiDAR-only baseline (using the same augmentation strategies as P3Former adopts), IAL achieves a 5.3% improvement, primarily due to a 7.5% increase from thing classes, demonstrating the effectiveness of image assistance in detecting and recognizing objects. In Table 3, IAL also demonstrates superior performance, achieving the highest scores across most metrics on the nuScenes leaderboard. These outstanding results highlight the effectiveness of our modules for modality alignment and compensation.

**SemanticKITTI** presents a significant challenge due to its use of only two front-view cameras, limiting the availability of image features to support LiDAR. As shown in Table 4, despite these constraints, our IAL achieves a 4.1% improvement in PQ over the state-of-the-art multi-modal baseline LCPS, demonstrating the robustness of our method even under limited image supervision.

### 4.3. Ablation Studies

To validate the effectiveness of our proposed components, we conduct comprehensive ablation studies on the overall proposal framework in Table 5 and provide detailed analyses for each individual module in Table 6. All experiments are conducted on the nuScenes validation set using the same hyper-parameters for fair comparison.

As shown in Table 5, compared to the baseline that uses only basic point cloud transformations (row 1), PieAug improves PQ by 2.7%, benefiting from better input alignment and enriched scene context. Building on this, GTF further boosts PQ by 2.7% and RQ by 2.1%, demonstrating that

*Table 5.* Ablation study of the proposed modules in our framework. "PIE" denotes the PieAug module.

| PIE | GTF | PQG | PQ | PQ$^\dagger$ | RQ | SQ | mIoU |
|---|---|---|---|---|---|---|---|
| | | | 75.7 | 78.1 | 84.4 | 88.3 | 73.8 |
| ✓ | | | 78.4 | 81.0 | 86.9 | 90.0 | 78.2 |
| ✓ | ✓ | | 81.1 | 83.5 | 89.0 | 90.9 | 80.2 |
| ✓ | ✓ | ✓ | **82.3** | **84.7** | **89.7** | **91.5** | **80.6** |

*Table 6.* Ablation study of PQG module. "Geo.", "Tex.", and "NP." represent geometric prior, texture prior, and no-prior queries, respectively. We set the total number of queries to 256 for a fair comparison. In configurations combining prior (geometric or texture) and no-prior queries, 128 queries are allocated to each set.

| Geo. | Tex. | NP. | PQ | PQ$^{th}$ | PQ$^{st}$ | mIoU |
|---|---|---|---|---|---|---|
| | | ✓ | 81.2 | 83.8 | 76.8 | 79.8 |
| ✓ | | ✓ | 81.3 | 83.9 | 77.0 | 80.0 |
| | ✓ | ✓ | 81.1 | 83.4 | 77.2 | 80.0 |
| ✓ | ✓ | | 80.7 | 83.0 | 77.0 | 80.0 |
| ✓ | ✓ | ✓ | **82.3** | **85.3** | **77.3** | **80.6** |

*Table 7.* Comparison of augmentation strategies. "Img" indicates whether image-synchronized augmentation is applied.

| Method | Img | PQ | PQ$^\dagger$ | RQ | SQ | mIoU |
|---|---|---|---|---|---|---|
| PolarMix | | 80.3 | 82.8 | 87.9 | 91.0 | 78.6 |
| LaserMix | | 80.6 | 83.0 | 88.5 | 90.8 | 79.3 |
| PieAug (ours) | | 81.4 | 83.7 | 89.1 | 91.0 | 80.1 |
| **PieAug (ours)** | ✓ | **82.3** | **84.7** | **89.7** | **91.5** | **80.6** |

unified scale embedding and accurate projection enhance multi-modal representations. Finally, incorporating the PQG module brings an additional 1.2% gain in PQ, validating our hypothesis that initializing queries with modality priors leads to more precise object predictions than using purely learnable parameters.

The effectiveness of the PQG module is validated in Table 6. Rows 1–3 show comparable performance, suggesting that purely learnable queries tend to overfit easy or redundant samples, even when uni-modal priors are available. Row 4 shows a slight drop, likely due to an excessive number of prior-based queries exceeding the number of ground-truth instances, resulting in more false positives. In contrast, our design (row 5) assigns strong geometric and texture priors to confident regions, while reserving learnable queries for harder, low-prior cases. This balanced allocation improves the model's ability to handle both easy and difficult samples, leading to superior overall performance.

### 4.4. Augmentation Methods Comparison

As shown in Table 7, we compare PieAug with LiDAR-only augmentation methods, including PolarMix (instance pasting and scene mixing) and LaserMix (inclination angle splitting) by 2.0% and 1.7% in PQ. Even with LiDAR-only augmentation, PieAug achieves superior performance, demonstrating its effectiveness as a generalized framework.

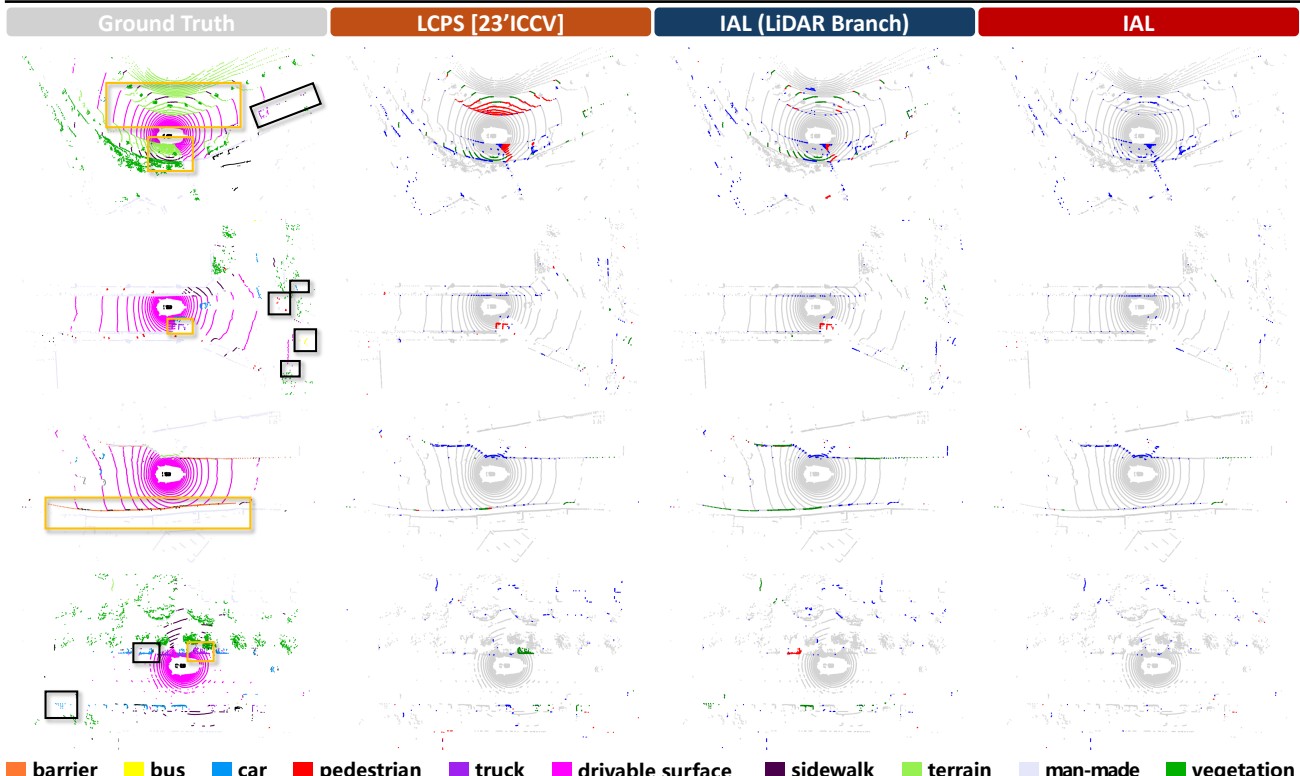

| Ground Truth | LCPS [23'ICCV] | IAL (LiDAR Branch) | IAL |

**■** barrier  **■** bus  **■** car  **■** pedestrian  **■** truck  **■** drivable surface  **■** sidewalk  **■** terrain  **■** man-made  **■** vegetation

*Figure 4.* Qualitative comparison of our method with the preliminary multi-modal panoptic segmentation baseline, LCPS. To highlight the differences, we mark **false positive** and **false negative** predictions, which affect recognition quality, as well as **well-matched** and **mismatch points** in true positive predictions, which impact segmentation quality. GT is colorized by semantic label. Best viewed in color.

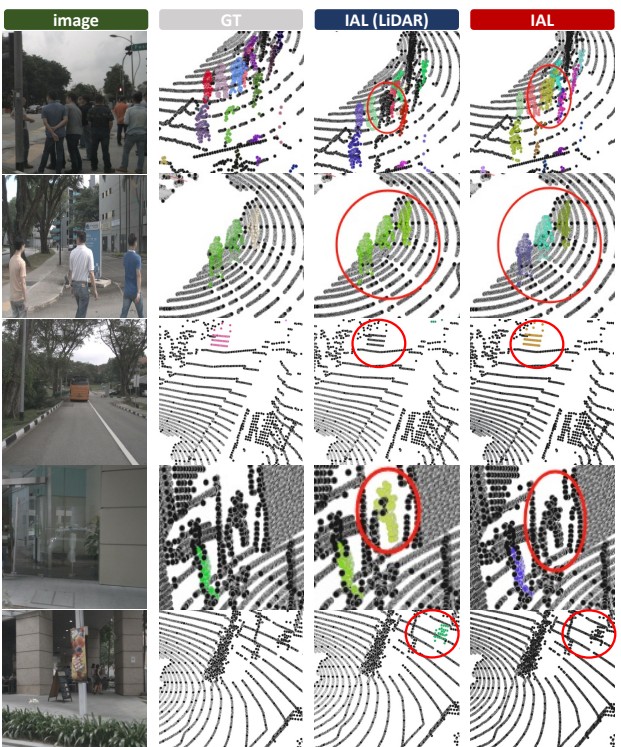

*Figure 5.* Visualization of instance predictions. Red circles highlight instances where the LiDAR branch fails to segment correctly, but our multi-modal method succeeds. Best viewed in color.

### 4.5. Qualitative Results and Discussion

We present qualitative evaluations on nuScenes validation set. As illustrated in the error maps in Fig. 4, our method notably reduces false positives (red points) and false negatives (green points) compared to LCPS. Furthermore, IAL outperforms its LiDAR branch in detecting remote objects (highlighted in the black boxes) and recognizing ambiguous classes (in yellow boxes), leveraging the assistance of image data. In Fig. 5, we compare instance prediction with GT, LiDAR branch, and IAL alongside the corresponding images. IAL showcases significant performance improvements in: (1) distinguishing multiple objects when they are clustered together (rows 1 and 2); (2) detecting distant objects (row 3); (3) recognizing false positive objects (rows 4 and 5).

### 5. Conclusion

This paper proposes IAL, a multi-modal 3D panoptic segmentation framework that harmonizes LiDAR and images through PieAug (synchronized augmentation), GTF (geometry-guided fusion), and PQG (prior-based queries). IAL directly predicts panoptic results via a transformer decoder, eliminating post-processing and achieving state-of-the-art performance on nuScenes (82.3% PQ) and SemanticKITTI (63.1% PQ). Texture-prior queries enhance small/distant object recognition, while geometric-prior queries improve large/nearby instance localization.

## Impact Statement

This paper aims to enhance multi-modal 3D panoptic segmentation. There are minor potential societal consequences of our work, none of which we feel must be specifically highlighted here.

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

## Supplementary Materials

The supplementary materials are organized as follows: Sec. A extends the ablation study with token fusion stage; Sec. B provides additional qualitative results to visualize the improvements brought by each module; Sec. C analyzes the efficiency of IAL; Sec. D investigates how image inputs enhance LiDAR under perturbations from lighting and weather conditions; Sec. E discusses the potential broader impact of IAL; and Sec. F outlines current limitations and future work.

## A. Ablation Study of Token Fusion

To analyze the effectiveness of the Geometric-guided Token Fusion (GTF) module, we divide GTF into two components: Token Selection (Sel) and Token Positional Embedding (PE). The full version of GTF uses all physical points within a cylindrical voxel (denoted as "set") for token selection and embeds the scale between all extreme points ("scl") to indicate perception regions. Alternative designs degrade token selection to a virtual center ("ctr") and positional embedding to the center or extreme points ("ext") of the voxel. All results are evaluated using the same experimental setting as the ablation studies in the main manuscript.

Table 8 reveals that using a point set rather than the virtual center for every token to construct the image feature contributes up to a 0.9% increase in PQ performance, and an up to 0.7% improvement in RQ. This verifies that precise LiDAR-image projection helps LiDAR voxels find corresponding image patches, and image features assist in recognition. For positional embedding, using scale-aware embedding indicates the potential perception regions of both LiDAR and image tokens, achieving the highest performance. This advanced improvement diminishes when scaling is degraded to using extreme points or solely the center of the voxel.

*Table 8.* Ablation study of the GTF module. "Sel" and "PE" denote the designs for token selection and positional embedding, respectively. We evaluate different configurations for component ablation: "ctr" represents the voxel virtual center, "set" refers to physical points within a voxel, and "-", "ctr", "ext", and "scl" indicate not implementing PE, the use of center, extreme, or scale embeddings.

| Sel | PE | PQ | PQ$^\dagger$ | RQ | SQ | mIoU |
|-----|-----|------|------|------|------|------|
| ctr | – | 78.4 | 81.0 | 86.9 | 90.0 | 78.2 |
| ctr | ctr | 79.7 | 82.2 | 87.8 | 90.5 | 78.3 |
| ctr | ext | 79.7 | 82.3 | 87.7 | 90.6 | 78.5 |
| ctr | scl | 80.4 | 82.7 | 88.3 | 90.7 | 78.4 |
| set | – | 79.3 | 81.7 | 87.5 | 90.3 | 77.7 |
| set | ctr | 80.6 | 83.0 | 88.4 | 90.8 | 80.1 |
| set | ext | 80.0 | 82.4 | 88.0 | 90.6 | 77.9 |
| set | scl | **81.1** | **83.5** | **89.0** | **90.9** | **80.2** |

## B. Qualitative Results for Modular Performance

We present qualitative evaluations of the ablation studies for the GTF and PQG modules on the nuScenes validation set, as illustrated in the error maps in Fig. 6. With the assistance of the GTF module (comparing column 3 with column 2), our model demonstrates improved performance in distinguishing ambiguous objects, such as recognizing barriers and pedestrians from the background (highlighted in yellow boxes in rows 2 and 5), as well as differentiating buses and trucks from cars (rows 3-5). These classes often share similar geometric appearances, especially when point clouds are sparse. However, images provide rich texture features that help distinguish each class, even in limited regions. The GTF module enhances LiDAR voxel data by embedding it with more accurate image features, allowing for better receptive field estimation for each cylindrical voxel and facilitating a strong alignment of LiDAR and image features. The PQG module further enhances object perception, as shown by the comparison between column 4 and column 3, especially for small-scale and remote objects highlighted in black boxes. Even for small objects like pedestrians, bicycles, cars, and trailers, which have few points, the PQG module succeeds in accurate detection, demonstrating its superior performance by utilizing both geometric and texture priors, as well as the learnable capability of no-prior queries.

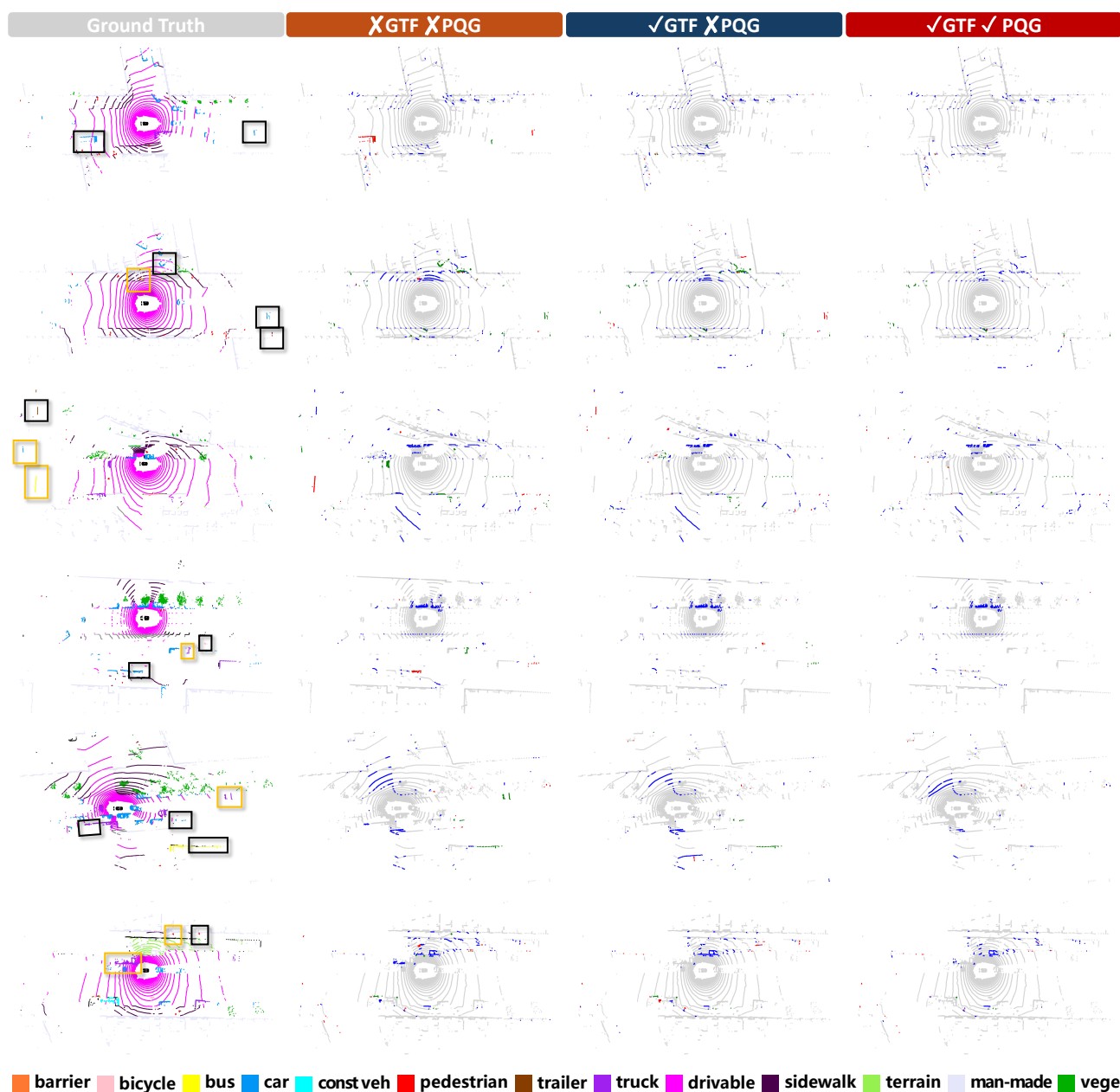

Figure 6. Qualitative comparison of the ablation study for GTF and PQG modules. To emphasize the differences, we mark **false positive** and **false negative** predictions, which affect recognition quality, as well as **well-matched** and **mismatch points** in true positive predictions, which impact segmentation quality. GT is colorized by semantic label. Best viewed in color.

## C. Time and Memory Cost

We compare inference speed, model size, and performance between our method and the main baseline, LCPS. We also report a lightweight variant (denoted by *), which excludes the 2D mask pre-processing step (Grounding-DINO and SAM) to highlight the efficiency of our framework's major components. All latency measurements are conducted on a single NVIDIA A40 GPU with batch size 1. For a fair comparison, we measure LCPS latency using its official codebase on our hardware. As shown in Table 9, our method achieves over 2× faster inference and a +2.5% gain in PQ compared to LCPS. Even when including the mask generation time, our approach remains comparable in speed.

*Table 9.* Comparison of models in terms of inference speed (FPS), model size (#Params), and Panoptic Quality (PQ). * denotes the result of core components of our model. All latency measurements are conducted on the same device.

| Model | FPS | #Params (M) | PQ |
|---|---|---|---|
| LCPS | 1.7 | 77.7 | 79.8 |
| IAL | 4.0* | 81.8* | 82.3 |
| | 0.9 | 859.9 | 82.3 |

*Table 10.* Performance on the full nuScenes validation set and its nighttime/rain subsets. Best results are highlighted in **bold**.

| Model | Full Val Set | Night Split | Rain Split |
|---|---|---|---|
| # of scan | 6,019 | 602 | 1,088 |
| LCPS | 79.8 | 64.3 | 76.8 |
| IAL (LiDAR branch) | 77.0 | 63.2 | 73.1 |
| IAL (full model) | **82.3** | **70.5** | **81.2** |

## D. Image Assists LiDAR Under Adverse Conditions

We evaluate the performance of our IAL on the **nighttime** and **rain** splits of the nuScenes val set. In the nighttime scenario, image quality is significantly degraded; in the rain scenario, both LiDAR and image encounter perturbations. As shown in Table 10, IAL (row 4) outperforms LCPS (row 2) not only on the full set but also under each adverse condition. We further ablate cross-modal interaction by comparing the full IAL to its LiDAR-only branch (row 3). Even under the degraded nighttime and rain splits, the full model gains +7.3% and +8.1% on PQ, respectively, confirming the image's effective assistive role for LiDAR. This improvement can be attributed to two main factors: 1. Modality-synchronized augmentation (PieAug), which exposes the model to more diverse samples, including nighttime and rain scenarios, by mixing synchronized LiDAR and image data. This allows the model to generalize better to rare conditions like nighttime scenes. 2. The combination of three types of queries in our PQG module, where no-prior queries complement the texture-prior and geometric-prior queries, helping the model to effectively identify potential instances. Additionally, pre-trained Grounding-DINO and SAM further stabilise 2D mask generation under distribution shifts thanks to their large-scale training.

## E. Potential Broader Impact of This Work

Based on the details in the paper, the broader impacts of this work can be highlighted as follows:

- The proposed multi-modal 3D panoptic segmentation framework (IAL) advances the field of autonomous driving by improving object detection and segmentation through the integration of LiDAR and image data. This technology has direct implications for the safety, accuracy, and efficiency of autonomous vehicles, particularly in complex, real-world environments. By addressing challenges such as the sparsity of LiDAR data and the difficulty of recognizing small or distant objects, this work enhances perception systems, enabling more reliable decision-making in autonomous driving.

- Furthermore, the advancements in modality-synchronized augmentation (PieAug) and geometric-guided token fusion (GTF) represent significant contributions to the broader field of sensor fusion in robotics and autonomous systems. These innovations could be adapted for other applications requiring high-precision environmental understanding, such as robotics, agriculture, and urban planning.

- As with all AI technologies, ethical considerations should be taken into account, particularly concerning privacy, security, and the potential for job displacement in industries such as transportation and logistics. However, the broader societal impact is generally positive, particularly in terms of enhancing public safety and reducing the risks associated with human error in driving.

# F. Limitations

While our work achieves strong results across key metrics, it is important to acknowledge certain limitations. Specifically, the sampling method in our Query Initialization (PQG) module relies on a relatively simple and generic approach. While our work demonstrates strong performance across key metrics, it is important to note that the extraction of texture-prior queries relies on generic, large-scale pre-trained models rather than methods specifically designed for this task or benchmark. Although this approach ensures broad applicability, it may not fully leverage task-specific characteristics that could further enhance performance. Nevertheless, through our carefully designed PQG module, we still achieve competitive results. In future work, we plan to explore more specialized sampling methods tailored to the task, which could further improve the quality of texture-prior queries and overall performance.

