# OpenReview forum: "How Do Images Align and Complement LiDAR? Towards a Harmonized Multi-modal 3D Panoptic Segmentation"
_ICML.cc/2025/Conference — ICML 2025 poster_

### Official Review · Reviewer_DcuJ · 2025-03-10

**Overall Recommendation:** 2

**Summary:**

This paper presents Image-Assists-LiDAR (IAL), a new panoptic 3D multi-modal segmentation method combining camera images with LiDAR point clouds. The key contributions are PieAug, multi-modal data augmentation with aligned data, Geometric-guided Token Fusion (GTF) for effective cross-modal fusion of features, and Prior-based Query Generation (PQG) with geometric (LiDAR) priors in a transformer decoder, with texture (image) priors. IAL achieves state-of-the-art results for nuScenes (PQ=82.0%) and SemanticKITTI (PQ=63.1%)

**Claims And Evidence:**

They are generally backed by extensive experiments and ablations. The significant improvements in PQ in typical benchmarks supports the performance of each of the proposed modules.

**Essential References Not Discussed:**

The work does not mention comparisons or discussions with other top-performing LiDAR-only methods including LidarMultiNet [AAAI'23](PQ~81.4%, mIoU~82.2% on nuScenes test set) and PUPS [AAAI'23] (PQ~64.4% on SemanticKITTI validation set). A mention of those will give a better context to the performance gains and clearly highlight the strength of their multi-modal approach.

**Experimental Designs Or Analyses:**

Experimental design is solid and rigorous. Ablation experiments nicely demonstrate the contribution of each component. Authors should nevertheless indicate whether performance improvements are primarily the result of new fusion strategies or the application of pre-trained strong image models (SAM, GroundingDINO).

**Methods And Evaluation Criteria:**

The chosen methods and datasets (nuScenes and SemanticKITTI) are apt and standard for the evaluation of panoptic segmentations in the 3D domain. The proposed modules (PieAug, GTF, PQG) are well-motivated, addressing apparent limitations in the prior work.

**Other Comments Or Suggestions:**

Minor improvements could include explicitly comparing inference speed and model size against baseline models.

**Other Strengths And Weaknesses:**

Strength
Transparent methodological developments (PieAug, GTF, PQG)
Strong experimental validation with comprehensive ablations.
Clear text with good figures presented.
Open Source Code Release Commitment

Weakness
Exclusion of high-performance LiDAR-only methods (for example, PUPS, LidarMultiNet) from comparison
There is no mention of the efficiency of inferences or real-world runtimes of heavy external image models (SAM, GroundingDINO).

**Questions For Authors:**

1. How quickly is IAL’s inference compared to its complexity and reliance on external models like SAM? Could it be run in real-time?

2. How significant is the contribution of the use of strong pre-trained image models compared to the new fusion strategy? Would less complex models be able to produce comparable results?

3. How robust is IAL when a modality is highly degraded (i.e., low image quality at night)? Does it degrade gracefully to the performance of LiDAR alone?

**Relation To Broader Scientific Literature:**

The paper correctly positions itself in the context of the most up-to-date multi-modal panoptic segmentation techniques (Panoptic-FusionNet, LCPS). It correctly identifies the limitations of the state-of-the-art approaches (post-processing-intensive, misaligned augmentation) and correctly positions its contributions (aligned augmentation, fusion using transformers) in the context of the state-of-the-art.

**Theoretical Claims:**

There were no proofs given or required.

---

> ### Author Rebuttal · Authors · 2025-04-01
>
> We sincerely appreciate the reviewer’s thoughtful and valuable feedback.
>
> ### C1: Contribution of performance improvement
>
> As in Table 8 of our supplementary material, using only texture-prior and no-prior queries (row 3) performs worse than the combination of all three query types (last row). This indicates that the performance improvements primarily stem from our novel fusion strategy—the combination of 3 types of queries in the PQG module—rather than solely relying on pre-trained strong image models, which only influence texture-prior query generation.
>
>
> ### C2: Discussions with other top-performing LiDAR-only methods
>
> Thank you for suggesting these two references. We will include them in the final version and provide the following discussion:
>
> **LidarMultiNet**: IAL outperforms LidarMultiNet on **all key panoptic segmentation metrics** despite using only a single frame and standard annotations, whereas LidarMultiNet leverages **extra** temporal frames, detection annotations, and multi-task learning. While LidarMultiNet slightly surpasses IAL in mIoU, this metric primarily reflects 3D semantic segmentation and is not the main evaluation metric for 3D panoptic segmentation. Moreover, some of LidarMultiNet’s effective components, such as GCP and two-stage, **do not conflict with** our multimodal method and could potentially be integrated into IAL to further enhance multi-modal 3D panoptic segmentation performance.
>
> **PUPS**: PUPS exibits inconsistent performance across datasets. Specifically, it significantly outperforms the baseline method Panoptic-PHNet on SemanticKITTI (Table 1 & 2 of PUPS) but achieves only comparable performance to Panoptic-PHNet on nuScenes validation set (Table 3 of PUPS). In contrast, IAL consistently achieves superior result over Panoptic-PHNet on both datasets. Since PUPS has not released its code for reproducibility, we hypothesize that its inconsistent performance trend may stem from overfitting to the simpler SemanticKITTI. Compared to SemanticKITTI, nuScenes presents greater challenges due to its larger scale, more diverse domains.
> IAL attains the highest performance on nuScenes, further demonstrating its robustness and practical effectiveness.
>
>
>
> ### C3: Inference speed and model size comparison
>
> Good catch! We provide a comparison between across all methods in the table below.
>
> | Model               | Mask Proposal Method    | FPS  | Params (M)    | PQ   |
> |--|--|--|--|--|
> | LCPS                | -                       | 1.7  | 77.7     | 79.8 |
> | IAL         | Grounding DINO + SAM    | 0.9  | 859.9    | 82.3 |
> | IAL*         | Grounding DINO + SAM    | 4.0* | 81.8*    | 82.3 |
> | IAL          | Mask R-CNN              | 2.7  | 123.8M   | 81.7 |
>
> To isolate the impact of external image models (SAM, GroundingDINO), we exclude 2D preprocessing step and report the efficiency of the remaining pipeline in 3rd row (denoted by *). Notably, the inference speed of our core pipeline is faster than LCPS, showing the efficiency of our design.
>
> We replace the *heavy* 2D preprocessing step (GroundingDINO and SAM) with a *lightweight alternative*, Mask R-CNN (ResNet50 backbone), as reported in 4th row. This variant strikes a balance between speed and performance, running at 2.7 FPS (1.6× faster than LCPS) while maintaining 81.7 PQ. These results indicate that while external 2D mask generation models introduce additional computational costs, our **query generation design remains flexible**, accommodating different 2D mask proposal models to **balance high-quality performance and fast inference speed**. More importantly, our **core framework remains highly efficient and robust** to different 2D preprocessing choices. Also, further optimizations such as model pruning or quantization could further accelerate inference.
>
>
> ### C4: Robustness when a modality is degraded
>
> Great suggestion! We evaluate the performance of IAL by comparing the LiDAR-only branch with the full model on the **nighttime** split (602 scans) of the nuScenes val set (6,019 scans), where image quality is significantly degraded. The results, (63.2 vs 70.5 PQ), show that despite the low image quality, our IAL (full model) still achieves a **+7.3\% PQ gain** over the LiDAR-only model. This improvement can be attributed to two factors: 1. **Modality-synchronized augmentation (PieAug)**, which exposes the model to more diverse samples, including nighttime scenarios, by mixing synchronized LiDAR and image data. It allows the model to generalize better to rare conditions like night scenes. 2. The combination of **3 types of queries** in PQG, where no-prior queries complement the texture-prior and geometric-prior queries, helping the model to effectively identify potential instances. Additionally, pre-trained Grounding-DINO and SAM models contribute to enhancing the robustness of 2D mask generation in night conditions, as they have been trained on vast amounts of data, improving generalization to such challenging scenarios.

---

> > ### Comment · Reviewer_DcuJ · 2025-04-02
> >
> > Thanks for your detailed rebuttal and for addressing several of my points clearly. However, the original omission of comparisons and discussions with other top-performing methods and the high complexity and practical deployment challenges still weakens the paper’s context regarding significance. Based on the overall originality and significance, I maintain my original score.

---

> > > ### Author Response · Authors · 2025-04-04
> > >
> > > Again, we sincerely thank Reviewer #DcuJ for carefully reading and acknowledging our rebuttal. We truly appreciate your continued engagement and the recognition that some of your concerns have been addressed. Below, we would like to further clarify two points to directly address your remaining concerns.
> > > ### R1: No Comparison with LidarMultiNet and PUPS in the Original Submission
> > >
> > > For ​**​LidarMultiNet​**​, we did not initially include it because it adopts a ​**​different experimental setting​**​ from ours and other baselines. Specifically, it utilizes additional temporal frames, detection annotations, and multi-task learning, making direct comparison challenging. Furthermore, LidarMultiNet is ​**​not open-sourced​**​, preventing a fair reproduction of its results for evaluation under the common setup.
> > >
> > > Regarding ​**​PUPS​**​, as explained in our rebuttal, it exhibits ​**​inconsistent performance trends​**​ across datasets. While it significantly outperforms Panoptic-PHNet (CVPR'22) on SemanticKITTI, it only performs comparably to Panoptic-PHNet on nuScenes, which raises concerns about its generalizability. Additionally, PUPS does ​**​not release official code​**​, making it infeasible to reproduce its results for a fair evaluation.
> > >
> > > For a clear comparison, we provide comparison results on the nuScenes and SemanticKITTI in Table 1. Only val set used due to the inconsistent LidarMultiNet's test result in paper vs. leaderboard. Importantly, on nuScenes, a more challenging benchmark for multi-modal 3D scene understanding, our method achieves significant improvements over all baselines, including LidarMultiNet and PUPS. We believe this strongly validates the effectiveness of our approach.
> > >
> > > Nevertheless, we acknowledge the importance of providing a broader context and will include a discussion of these two works in our final version.
> > >
> > > ||nuScenes|||||SemanticKITTI|||||
> > > |-|:-:|-|-|-|-|:-:|-|-|-|-|
> > > |Method|PQ|PQ+|RQ|SQ|mIoU|PQ|PQ+|RQ|SQ|mIoU|
> > > |LiDARMultiNet|81.8|-|89.7|90.8|82.0|-|-|-|-|-|
> > > |PUPS|74.7|77.3|83.3|89.4|-|64.4|68.6|74.1|81.5|-|
> > > |IAL|82.3|84.7|89.7|91.5|80.6|63.1|66.3|72.9|81.4|66.0|
> > >
> > > ​*Table 1: Method Comparison. "–" denotes results not available (no code)​*
> > >
> > > ### R2: High Complexity and Practical Deployment Challenges
> > >
> > > We appreciate your feedback and understand your concerns. However, we emphasize that our method is both complexity-flexible and robust, making it practical for real-world deployment.
> > >
> > > |Row|Model|Mask/Box Proposal|FPS|Params(M)|PQ (%)|
> > > |-|-|-|-|-|-|
> > > |1|LCPS|– |1.7|77.7 |79.8|
> > > |2|IAL|Grounding DINO + SAM|4.0*|81.8*|82.3|
> > > |3|IAL|Grounding DINO + SAM|0.9|859.9|82.3|
> > > |4|IAL-V1|HTC|2.4|218.4|81.9|
> > > |5|IAL-V2|Mask R-CNN|2.7|123.8|81.7|
> > > |6|IAL-V3|Grounding DINO 1.5 Edge|3.8|– |81.3|
> > >
> > > ​*​Table 2: Model Performance Comparison​*​
> > >
> > > First, as outlined in our rebuttal, our approach allows for flexible adaptation of the 2D mask generation (preprocessing) module, enabling deployment under different complexity and efficiency constraints. As shown in Table 2, the core design of our method (2nd row) is both highly effective and efficient compared to the main baseline, LCPS (1st row). Additionally, we introduce two more alternatives for the 2D mask generation module, HTC [1] and Grounding DINO 1.5 Edge [2], as shown in 4th and 6th rows. The results highlight the trade-off between accuracy and efficiency: when prioritizing accuracy, we can use Grounding-DINO and SAM (3rd row) - a more complex yet powerful module - to achieve superior performance (+2.5% PQ over LCPS), albeit with a lower inference speed. On the other hand, when efficiency is prioritized, adopting a lighter 2D mask generation model, such as Grounding DINO 1.5 Edge, yields higher FPS (>2x faster than LCPS) while still maintaining performance improvements over LCPS. Notably, across all 2D mask generation choices (rows 3 to 6), our IAL consistently outperforms LCPS, demonstrating the robustness and adaptability of our approach.
> > > ​
> > > |Model|Full Val Set|Night Split|Rain Split|
> > > |-|-|-|-|
> > > |# of scan|6019|602|1088|
> > > |LCPS|79.8|64.3|76.8|
> > > |IAL|​**​82.3​**|​**​70.5​**​| ​**​81.2​**|
> > >
> > > *​Table 3: Robustness Evaluation Under Adverse Conditions​*
> > >
> > > Second, our method maintains strong performance across various environmental conditions. Beyond the nighttime evaluation included in our rebuttal, we further assess our method under another challenging scenario, the "rain" split, compared to the main baseline, LCPS. As shown in Table 3, our IAL significantly outperforms LCPS in these adverse conditions, reinforcing its robustness and practical applicability.
> > >
> > > [1]Hybrid Task Cascade for Instance Segmentation. CVPR19
> > >
> > > [2]Grounding DINO 1.5: Advance the Edge of Open-Set Object Detection. arXiv24
> > >
> > > **We hope these additional experiments address your concerns and further demonstrate the significance of our contributions.**

---

### Official Review · Reviewer_qZ7o · 2025-03-10

**Overall Recommendation:** 4

**Summary:**

Aiming at 3D panoptic segmentation, the paper proposes using a transformer decoder to directly predict class labels and mask outputs. The authors further introduce a Geometric-guided Token Fusion (GTF) module and a Prior-based Query Generation (PQG) module to obtain effective queries and fuse tokens as input. In addition, a multi-modal data augmentation strategy is designed to augment both 3D and 2D inputs. The experiments show the effectiveness of the proposed method.

**Claims And Evidence:**

The claims are reasonable and supported. PieAug augments both modalities while maintaining synchronization across them. The GTF module addresses mismatches between modalities and scale issues by mapping the eight corner points during fusion, and the PQG module leverages priors from both modalities to initialize queries for better localization and performance.

**Essential References Not Discussed:**

The paper should also reference existing fusion modules proposed in multi-modal 3D segmentation literature, such as
+ ICCV 2023, UniSeg: A Unified Multi-modal LiDAR Segmentation Network and the OpenPCSeg codebase
+ ICLR 2025, Multimodality Helps Few-shot 3D Point Cloud Semantic Segmentation

**Experimental Designs Or Analyses:**

The experiments are conducted on two standard outdoor datasets, demonstrating the effectiveness of the proposed method. However, the ablation study is somewhat limited. For example, GTF requires projecting all the corner points for computing positional encoding and token fusion, and PQG uses three sets of queries, with two query groups needing external proposal models for initialization. It would be beneficial to know the computational efficiency of the method and the parameter count compared to previous methods.

**Methods And Evaluation Criteria:**

The proposed designs are logical, and the chosen evaluation benchmarks are suitable for the task.

**Other Comments Or Suggestions:**

Please see the Questions.

**Other Strengths And Weaknesses:**

The paper is well-written and easy to follow. The motivations for the design choices are clear and reasonable.

**Questions For Authors:**

GTF requires projecting all the corner points for computing positional encoding and token fusion, and PQG uses three sets of queries, with two query groups needing external proposal models for initialization. Could the authors provide the efficiency analysis such as comparing the computational cost and the parameter count with previous methods?

**Relation To Broader Scientific Literature:**

The proposed designs complement the broader literature and offer new insights.

**Theoretical Claims:**

There are no theoretical claims.

---

> ### Author Rebuttal · Authors · 2025-04-01
>
> We sincerely thank the reviewer for the positive feedback and constructive suggestions.
>
> ### C1: Suggestion for adding additional references.
> Thank you for pointing out these two papers. We will add them in the final version and provide the following discussions:
>
> 1. **UniSeg (ICCV’23)** proposes a unified multi-modal fusion strategy for LiDAR segmentation, focusing on fusing voxel-view and range-view LiDAR features with image features. However, UniSeg primarily targets semantic segmentation and does not explore how to utilize multimodal information for instance segmentation, which focuses on thing classes. In contrast, our design of *prior-based query generation* is specifically tailored for 3D instance segmentation. Furthermore, while UniSeg employs Cartesian voxelization, our method uses *cylindrical voxelization*, which has been proven more effective for panoptic segmentation in prior works like PolarNet (CVPR'20) and Cylinder3D (CVPR'21). We also incorporate *Geometric-guided Token Fusion* (including scale-aware positional encoding) to better accommodate this cylindrical representation.
>
> 2. **MM-FSS (ICLR’25)** focuses on few-shot 3D semantic segmentation in indoor scenarios by leveraging both explicit textual modalities and implicit 2D modalities for cross-dataset adaptation. This differs significantly from our work, which focuses on fully-supervised multi-modal 3D panoptic segmentation in outdoor scenarios. Additionally, the fused modalities in MM-FSS (text, image, and point cloud) differ from our approach, which uses multi-view images and LiDAR point clouds.
>
>
> ###  C2: Could the authors provide the efficiency analysis such as comparing the computational cost and the parameter count with previous methods?
>
> Thank you for your suggestion! We provide a comparison of inference speed (FPS), parameter count, and performance (PQ, the primary evaluation metric) between our method and LCPS, the main baseline, as shown in the table below.
>
> | Model          | Mask Proposal Method    | FPS  | Params   | PQ   |
> |--              |--                       |--    |--        |--   |
> | LCPS           | -                       | 1.7  | 77.7M    | 79.8 |
> | IAL (ours)     | Grounding DINO + SAM    | 0.9  | 859.9M   | 82.3 |
> | IAL (ours)*    | Grounding DINO + SAM    | 4.0* | 81.8M*   | 82.3 |
> | IAL (ours)     | Mask R-CNN              | 2.7  | 123.8M   | 81.7 |
>
>
> To better understand the computational cost and model size of our framework’s major components, we report a lightweight variant of our model (denoted by \* in the 3rd row), which excludes the 2D preprocessing step (Grounding-DINO and SAM). This analysis reveals that the main computational cost and parameter count stem from the sophisticated 2D preprocessing step. However, our core components - including GTF, geometric-prior query generation, 2D and 3D encoder, and transformer decoder - are highly efficient, achieving 4.0 FPS with only 81.8M of parameters.
>
> To further evaluate efficiency, we replace Grounding-DINO and SAM with Mask R-CNN (ResNet50 backbone), a lightweight 2D mask generation alternative, as reported in the 4th row. This variant achieves a well-balanced trade-off between speed and performance, with the full pipeline running at 2.7 FPS (1.6× faster than LCPS) while maintaining 81.7 PQ (+1.9% over LCPS). These results demonstrate that while external proposal models contribute to computational cost, our core method remains highly efficient and adaptable to different 2D preprocessing choices.

---

> > ### Comment · Reviewer_qZ7o · 2025-04-03
> >
> > My concerns have been addressed by the authors' detailed rebuttal. I think it meets the acceptance threshold and I have increased my score to recommend its acceptance.

---

> > > ### Author Response · Authors · 2025-04-05
> > >
> > > We sincerely thank you for your time, thoughtful feedback, and updated recommendation. We’re glad to hear that our responses have addressed your concerns, and we deeply appreciate your recognition of the value of our work. We are especially grateful for your positive assessment that our “proposed designs (PieAug, GTF, and PQG) are reasonable” with “clear motivation”, that their “effectiveness is supported” by the experiments, and that “they can complement the broader literature and offer new insights.” Your insightful comments have played a key role in strengthening our paper, and we truly appreciate your valuable contribution to its improvement.

---

### Official Review · Reviewer_CFKM · 2025-03-14

**Overall Recommendation:** 4

**Summary:**

This work proposed a novel framework for multi-model 3D panoptic segmentation. By leveraging the proposed modality-synchronized data augmentation (PieAug) with geometric-guided token fusion (GTF) and prior-based query generation (PQG), this work achieves new state-of-the-art (SOTA) performance on two challenging benchmarks, i.e. nuScenes and semantic-KITTI.

**Claims And Evidence:**

The authors claim the proposed PieAug can align multi-camera augmentation with corresponding LiDAR cylindrical voxels. This is proved by eq. 1 - 5 (theoretically), Fig. 3 (qualitatively), and Tab. 5 & 6 (quantitatively).

Authors also claim that GTF can better align LiDAR and multi-camera image features by eliminating projection errors and obtaining sufficient fields to fully utilize dense image features. This is supported by eq. 6 - 8 plus Fig. 2 (theoretically) and Tab. 5 (quantitatively).

The final claim is that PQG can provide sufficient proposals for panoptic segmentation using all kinds of priors. This is proved by Tab. 1 (theoretically) and Tab. 5 (quantitatively).

**Essential References Not Discussed:**

N/A

**Experimental Designs Or Analyses:**

The experimental designs are correct and solid. The analyses are sufficient and insightful both in the main paper and supplementary material.

**Methods And Evaluation Criteria:**

The methods are novel and interesting. Extensive experimental results support the claims and demonstrate its value for multi-modal 3D panoptic segmentation tasks. The evaluation criteria are standard.

**Other Comments Or Suggestions:**

- It will be nice to have a qualitative comparison among the ablation studies so that readers can understand how much the PieAug, GTF, and PQG affect the prediction.
- Tab. 8 is important for the readers to know how different prior queries affect the final results. It might be better to move it in the main paper.

**Other Strengths And Weaknesses:**

Strength:
- The proposed PieAug is interesting and seems quite effective. It solves the misalignment between modalities which is quite important and overlooked in the previous works.
- The proposed GTF is novel and meaningful. It provides a better way to align multi-model features and can be used in the popular transformer architectures.
- PQG is also novel and interesting, which derives from the observation of using ground truth (GT) center position. This finding is quite valuable to 3D panoptic segmentation.

Weakness
- Fig. 3 is too far from the Sec. 3.1, where the PieAug is introduced. It will be much better for the reader to get the full pictures of PieAug without page jumps and affect the reading.
- The impact statement seems to be on Page 9. I believe it should be in the supplementary material, or the overall contents management needs to be more careful.

**Questions For Authors:**

- Will the effectiveness of texture queries change a lot when changing the mask prior? For example, replacing Grounding SAM with Mask DINO or Mask R-CNN?

**Relation To Broader Scientific Literature:**

I believe the key contributions of the paper are related to the broader scientific literature, especially the proposed GTF. This interesting multi-modal features fusion will be quite valuable for other 3D perception tasks that are using multiple sensors.

**Theoretical Claims:**

As stated in the claims and evidence, the theoretical claims are correct and supported by quantitative results.

---

> ### Author Rebuttal · Authors · 2025-04-01
>
> We sincerely thank the reviewer for their constructive feedback and positive comments on our contributions.
>
> ### C1: Fig. 3 is too far from the Sec. 3.1
> Thank you for the suggestion. We will update the placement of Fig. 3 in the final version to ensure that PieAug is introduced and illustrated in a more cohesive manner.
>
> ### C2: The impact statement on page 9 should be in the supplementary material
> Thank you for the suggestion. However, we followed the official guidelines, which recommend placing the impact statement in a separate section at the end of the paper, co-located with the Acknowledgements, before the References.
>
> ### C3: Add qualitative comparison among ablation studies
> Great suggestion! Unfortunately, due to the constraints of the rebuttal policy, we are unable to include images or provide links to them on the OpenReview platform. However, we will add these qualitative comparisons in the final version of the paper.
>
> ### C4: Moving Tab. 8 to the Main Paper
> Thank you for your suggestion. We agree that Table 8 is crucial for showing how different prior queries impact the results. We will move it to the main paper in the final version.
>
> ### C5: Will the mask prior strongly impact texture queries' effectiveness? E.g., replacing Grounding SAM with Mask DINO or Mask R-CNN?
> Good catch! We have replaced the mask generation module (Grounding-DINO and SAM) with alternatives, including Mask R-CNN (as you suggested) and its follow-up, the HTC model [1], and report the results in the table below. As shown, the performance of our model remains robust to the choice of mask priors from different mask proposal methods, with only a minimal performance drop when using Mask R-CNN or HTC. This indicates that our texture-prior query generation design is flexible, accommodating different 2D mask proposal models, and can be integrated with advanced off-the-shelf methods (in the future) for further performance improvements.
>
> | **2D Mask Proposal**           | **PQ** |
> |---------------------------------|--------|
> | Mask R-CNN (ResNet50)           | 81.7   |
> | HTC (ResNeXt101)                | 81.9   |
> | Grounding-DINO + SAM            | 82.3   |
>
>
> [1] Chen Kai, et.al. Hybrid task cascade for instance segmentation. CVPR'19.

---

> > ### Comment · Reviewer_CFKM · 2025-04-08
> >
> > Thanks to the authors' efforts in the rebuttal. After checking other reviewers' feedback and authors' replies, I believe this work is valuable to 3D Panoptic Segmentation tasks and tend to accept it. I am willing to raise the score to 4 and hope the authors can revise the writing accordingly and release the code after acceptance.

---

> > > ### Author Response · Authors · 2025-04-09
> > >
> > > Dear reviewer,
> > >
> > > We sincerely thank you for your thoughtful feedback, the time you took to consider other reviewers' comments, your updated recommendation, and your in-depth engagement with our work. We’re glad that our responses have addressed your concerns and are especially grateful for your recognition of the value of our work.
> > >
> > > We appreciate your recognition that our proposed methods—PieAug, GTF, and PQG—are "*novel, interesting, and meaningful*", and that our claims are "*well-supported by sufficient evidence*" and "*solid experimental design*". We're particularly pleased that you found the GTF module to be a valuable contribution not only to our specific task but also to the broader field of multi-sensor 3D perception.
> > >
> > > We will further refine the paper based on your suggestions, and we will release our code to support the community. Thank you again for your constructive feedback and kind support.
> > >
> > > Best wishes,
> > >
> > > All authors

---

### Official Review · Reviewer_eBKA · 2025-03-17

**Overall Recommendation:** 3

**Summary:**

This paper proposes a new multi-modal framework for multi-modal 3D panoptic segmentation. First, the authors adopt a PieAug strategy to ensure consistency across LiDAR and image data during data augmentation. Then, they use a Geometric-Guided Token Fusion mechanism to fuse LiDAR and image tokens. Finally, several types of queries—geometric-prior queries, texture-prior queries, and no-prior queries—are used to obtain the final segmentation results. The authors conduct experiments on the nuScenes and Semantic KITTI datasets and demonstrate that the proposed method achieves better performance than state-of-the-art (SOTA) methods.

## update after rebuttal
The rebuttal address most of my concerns. I increase my rating to weak accept.

**Claims And Evidence:**

Yes.

**Essential References Not Discussed:**

No.

**Experimental Designs Or Analyses:**

Yes. I check the main experiments and the ablation studies.

**Methods And Evaluation Criteria:**

Yes. nuScenes and SemanticKITTI are popular used dataset in LiDAR panoptic segmentation tasks.

**Other Comments Or Suggestions:**

No.

**Other Strengths And Weaknesses:**

Strengths
1. The paper is well-written, with clear presentation and illustrations.
2. The authors provide detailed ablation studies to demonstrate the effectiveness of each component.
3. The authors conduct experiments on nuScenes and SemanticKITTI and achieve state-of-the-art (SOTA) performance.

Weaknesses

1. The adoption of cylindrical representation in the LiDAR branch lacks justification. There is no explanation as to why cylindrical representation is preferred over Cartesian representation. The scale ambiguity problem only exists in cylindrical representation. If Cartesian representation were used, the scale-aware position embedding might not be necessary, as the scale of each voxel would be consistent.
2. The overall pipeline is computationally expensive. The framework incorporates additional image models, such as Grounding-DINO and SAN, to extract 2D masks and use them as texture-prior queries. Comparing the proposed method with other methods without considering FLOPs or latency is unfair.
3. The idea of adopting proposals from heatmaps (e.g., TransFusion) or image modalities (e.g., Fully Sparse Fusion ) as priors is not novel. As shown in the supplementary materials, combining texture priors with geometric priors does not achieve better performance. The authors hypothesize that this is due to an excessive number of prior-based queries. However, adding no-prior queries increases the total number of queries but results in better performance. The authors should provide a more convincing explanation for this observation.

**Questions For Authors:**

Please refer to the Weaknesses in Other Strengths And Weaknesses.

**Relation To Broader Scientific Literature:**

1. Prior methods also adopt prior queries\proposals from LiDAR or Image modality to enhance the performances in LiDAR detection tasks, such as TransFusion and FSF. This paper adopts prior queries in the task of panoptic segmentation.
2. Prior methods adopt image painting, cross-attention, deformable attention to fuse multi-modal information. This paper adopt directly add fusion with scale aware positional embedding.

**Theoretical Claims:**

There are not theoretical claims in this paper.

---

> ### Author Rebuttal · Authors · 2025-04-01
>
> Thank you for your valuable feedback on the voxel representation, the model efficiency, and the design of query initiation. Regarding your questions, we provide the following detailed explanation.
>
> ### Q1: Justification of the adoption of cylindrical representation in the LiDAR branch.
> While Cartesian voxelization ensures uniform scales, it inherently struggles with the density imbalance in LiDAR point clouds. Due to **radial sparsity**—where points are denser near the sensor and sparser at a distance—Cartesian voxels fail to accommodate this variation, potentially causing uneven feature distribution and information loss, particularly in distant regions.
>
> The advantage of cylindrical over Cartesian voxel representations is well-documented in prior work, e.g., Cylinder3D (in CVPR’21, Sec. 3.2) and PolarNet (in CVPR’20, Sec 3.3), showing superior performance through adaptive partitioning of sparse distant points.
> Additionally, adopting cylindrical representation aligns with high-performing baselines like P3Former (IJCV’25) and LCPS (ICCV’23), ensuring consistency and fair comparison.
>
> ### C2: Computation cost comparison
> We compare inference speed (FPS), model size (Params), and performance (PQ, the primary evaluation metric) between our method and LCPS, the main baseline. Additionally, we report a lightweight variant of our model, denoted by *, which excludes the 2D preprocessing step (Grounding-DINO and SAM) to highlight the efficiency of our framework’s major components. Since the 2D preprocessing can be replaced with any off-the-shelf model, this variant demonstrates the trade-off between speed and performance. All latency measurements are conducted on a single NVIDIA A40 GPU with batch size 1. For fair comparison, we measure LCPS latency using its official codebase on our hardware.
>
> |        | Mask Proposal Method       | FPS   | Params (M)   | PQ  |
> |--|--|--|--|--|
> | LCPS        | -                  | 1.7   | 77.7    | 79.8|
> | IAL*   | Grounding DINO+SAM         | 4.0 | 81.8   | 82.3|
> | IAL   | Grounding DINO+SAM         | 0.9   | 859.9   | 82.3|
> | IAL   | Mask R-CNN                   | 2.7   | 123.8  | 81.7|
>
> As shown in the table above, with Grounding-DINO and SAM, IAL achieves slightly lower FPS than LCPS (0.9 vs. 1.7) but **significantly improves PQ** (82.3 vs. 79.8), making it ideal for accuracy-critical applications. Removing the 2D preprocessing step (row 2) makes our model 2.4× faster than LCPS. The choice of 2D preprocessing is flexible, balancing accuracy and efficiency. For instance, replacing Grounding-DINO and SAM with **Mask R-CNN** (ResNet50 backbone, IAL-MaskRCNN, row 4) yields 2.7 FPS (1.6× faster than LCPS) while maintaining a strong PQ of 81.7 (+1.9\% over LCPS).
>
> ### C3.1: Novelty of prior-based query generation
> We respectfully disagree with the comment. Our prior-based query generation (PQG) is both novel and well-motivated, as acknowledged by reviewer #DcuJ. PQG fundamentally differs from TransFusion and FSF in both design and query generation mechanism.
>
> From a design perspective, unlike TransFusion and FSF, which solely rely on prior-based queries, our no-prior query solution specifically targets objects missed by geometric (3D) and texture (2D) priors, a critical gap for cross-modal fusion that has been **unaddressed in prior works**.
>
> In terms of query generation, PQG introduces a **more advanced** approach that enhances prior utilization. While TransFusion fuses image features with BEV LiDAR features for heatmap prediction (which doesn't fully leverage the rich texture features and may harm each modality during fusion), our PQG module generates 2D and 3D prior-based queries independently.
> This allows us to fully leverage the rich texture information without information loss and perform late fusion, better complementing the two modalities. As for FSF, they do not explicitly use queries; instead, they extract potential instances from both 2D and 3D branches and merge them via self-attention for final prediction.
> Our 2D query generation differs in that we use clustering to group points within a frustum and then apply Farthest Point Sampling (FPS) to select potential candidates, effectively mitigating outliers (e.g., background points). In contrast, FSF assumes all points within a mask frustum belong to the same semantic object, which can cause semantic ambiguity.
>
> ### C3.2: Performance explanation when adding "no-prior queries"
> We would like to clarify that we **keep the total number of queries the same** (as stated in L616-617) across all settings in Table 8. This ensures that the observed differences in performance are not influenced by the total number of queries, but instead reflect the contribution of different query combinations. Thus, the performance improvement when adding no-prior queries cannot be attributed to an increased number of queries, but rather to the complementary benefits of no-prior queries in enhancing the model’s ability to handle missed objects.

---

### Decision · Program_Chairs · 2025-05-01

**Decision:**

Accept (poster)

**Comment:**

This paper presents a framework for multi-modal 3D panoptic segmentation by introducing PieAug, a modality-synchronized data augmentation strategy designed to ensure alignment between LiDAR and image inputs. Additionally, it proposes the GTF and PQG modules, which effectively combine image and LiDAR features as tokens and queries for the transformer decoder.

Reviewers raised concerns regarding the novelty of the work and sought clarification on aspects such as cylindrical representation, the prior-based query generation module, and computational costs.
The final ratings are mixed, weak accept, 2 accept, and weak reject.

After a thorough review, including consideration of both the authors' responses and the reviewers' comments, AC has decided to accept the paper. The authors must adhere to their commitments and rigorously revise the paper per their promises.